# Snakebite Treatment in Tanzania: Identifying Gaps in Community Practices and Hospital Resources

**DOI:** 10.3390/ijerph19084701

**Published:** 2022-04-13

**Authors:** Felicia Margono, Anne H. Outwater, Michael Lowery Wilson, Kim M. Howell, Till Bärnighausen

**Affiliations:** 1Heidelberg Institute for Global Health (HIGH), Heidelberg University, 69117 Heidelberg, Germany; michael.l.wilson@utu.fi (M.L.W.); till.baernighausen@uni-heidelberg.de (T.B.); 2School of Nursing, Muhimbili University of Health and Allied Sciences, Dar es Salaam P.O. Box 65001, Tanzania; anneoutwater@yahoo.com; 3Injury Epidemiology and Prevention Research Group, Turku Brain Injury Center, Division of Clinical Neurosciences, Turku University Hospital, University of Turku, 20700 Turku, Finland; 4Department of Zoology and Wildlife Conservation, University of Dar es Salaam, Dar es Salaam P.O. Box 35131, Tanzania; kimhowellkazi@gmail.com; 5Harvard Center for Population and Development Studies, Cambridge, MA 02138, USA

**Keywords:** snakebite, envenomation, treatment, Tanzania

## Abstract

Snakebite envenoming causes more than 140,000 deaths annually and at least triple this number of disabilities. The World Health Organization classified snakebite as a Neglected Tropical Disease in 2017 and developed a strategy to halve death and disability from snakebite by 2030. To achieve this goal, snakebite victims need to receive safe and effective treatment. This descriptive, cross-sectional study surveyed student health professionals (N = 312) in Dar es Salaam, Tanzania, and was designed to identify major gaps in community practices and hospital resources for snakebite treatment. Participants reported using traditional community practices (44%, 95% confidence interval (CI) = 39–50%), allopathic practices (7%, 95% CI = 5–11%), or a combination of both (49%, 95% CI = 43–54%) to treat snakebite. Harmful practices included tight arterial tourniquets (46%, 95% CI = 41–52%) and wound incisions (15%, 95% CI = 11–19%). Many participants (35%, 95% CI = 29–40%) also turned to traditional healers. Students who treated snakebite injuries within the last 5 years (N = 69) also reported their general experiences with snakebite in hospitals. Hospitals often lacked essential resources to treat snakebite victims, and 44% (95% CI = 30–59%) of snakebite victims arrived at a hospital only three or more hours after the bite. A significant percentage of snakebite victims experienced lasting damage (32%, 95% CI = 20–47%) or death (14%, 95% CI = 7–25%). Snakebite outcomes could likely be improved if hospitals were universally and consistently equipped with the essential resources to treat snakebite victims, such as antivenoms. Educational interventions aimed at communities should focus on discouraging tourniquet use and tampering with the wound. Collaboration between the allopathic and traditional health system could further boost snakebite outcomes because traditional healers are often the first health workers to see snakebite victims.

## 1. Introduction

Snakebite is a serious issue in tropical and subtropical regions worldwide, claiming more than 140,000 lives annually and causing three to four times the number of disabilities [1,2,3]. The World Health Organization (WHO) recognized snakebite as a Neglected Tropical Disease (NTD) in 2017 and in 2019 launched a global initiative that aimed to halve the numbers of deaths and cases of disability per annum by 2030 [4].

Harm is caused by snakebite envenoming (SBE), during which a snake injects several different types of toxins subcutaneously or intramuscularly after biting [5,6]. The three clinical toxins with the greatest impact in Tanzania are neurotoxins, which cause respiratory paralysis; cytotoxins, which cause swelling and tissue necrosis; and hematotoxins, which cause extensive bleeding and tissue destruction. All of these responses can result in death [6,7,8]. The most common chronic conditions resulting from poor treatment outcomes are musculoskeletal and severe cases involve tissue necrosis, which often necessitate amputations [9]. Snakebite disproportionately affects young subsistence workers in rural areas for whom disabilities resulting from snakebite are particularly damaging, because their livelihoods depend on their ability to do manual labor [10].

Mortality and disability caused by SBE disproportionally affect the world’s poorest people [11]. Moreover, the 43,000 cases of snakebite that are reported annually across western, eastern, central, and southern Africa are likely a substantial underestimate of the true count, because many snakebite cases are not reported [3]. Tanzania is one of the countries with very little data on the extent, nature, and consequences of SBE, even though it suffers from high numbers of envenoming and snakebite deaths each year [3]. Few countries are home to more medically relevant snake species than Tanzania, which has 22 species capable of causing morbidity, disability, and death [12].

Antivenom, also known as antivenin or antiserum, is currently the only specific, evidence-based treatment for SBE [13]. However, antivenoms may not be available in the hospitals where snakebite victims seek care [13]. Many snakebite victims also use traditional practices in lieu of or before going to hospitals [14] where antivenom is available because snakebite is often perceived as a supernatural event.

An understanding of snakebite treatment practices in local communities and the availability of specific hospital resources can guide interventions and policies to boost snakebite outcomes. This descriptive, cross-sectional study was designed to improve our understanding of community practices and hospital resources for snakebite treatment in Tanzania.

## 2. Materials and Methods

### 2.1. Study Design

We used a standardized questionnaire to interview nursing students at the Muhimbili University of Health and Allied Sciences (MUHAS) School of Nursing, in Dar es Salaam, Tanzania. The study was retrospective and primarily quantitative. The questionnaire consisted mainly of multiple-choice questions. Qualitative open-ended questions were used to verify and substantiate certain quantitative answers.

### 2.2. Data Collection

Data were collected from students enrolled at the MUHAS School of Nursing between 2014 and 2019. All students who were enrolled in a Bachelor of Science degree program in nursing, midwifery, or nursing management, or a master’s degree in midwifery or critical care and trauma were offered the opportunity to participate in our survey on snakebite. Our sample was intended to represent the life experiences of MUHAS nursing students with snakebite; nursing students at MUHAS are likely similar to nursing students in other parts of Tanzania, so that our results may also generalize well to the national population of nursing students. Prior to the survey, we briefed students on the purpose of the study and obtained consent for their participation. 

### 2.3. Participants

We divided the students’ academic programs into two categories: Bachelor of Science in Nursing and ‘other’. The former is a 4-year program with an additional year of internship, while the latter is a broad category encompassing degree programs such as the Bachelor of Science in Midwifery, Bachelor of Nursing Management, and Master of Science in Critical Care and Trauma. The ‘other’ category mostly consisted of mature students with working experience.

All students were asked to answer the qualitative question regarding snakebite treatment in their communities, whereas the quantitative and qualitative questions regarding hospital treatment were answered only by students with experience of treating snakebite injuries within the last 5 years. Only questionnaires accompanied by a signed consent form were used for the analyses.

### 2.4. Ethical Statement

The original questionnaire and data collection protocol were approved by the MUHAS Research and Publications Committee. Participants were identified by randomly generated serial numbers, and any names or contacts that may have been placed at the end of the questionnaire were voluntarily given. Participants had the right to withdraw at any time, and as a result there were no risks to the participants during the course of this study. Participants also benefited from the study: At the end of the session, using the guidelines in Appendix B, they were instructed on the proper care of snakebite victims.

### 2.5. Quantitative Analysis

Students were surveyed on their basic demographic characteristics, community practices, and the hospital resources available for snakebite treatment in hospitals, as well as snakebite patients’ treatment outcomes. The data were entered into a Microsoft Excel Version 16.59 table and cleaned, and RStudio Version 1.2.5033 was used to calculate descriptive summary statistics and visualize the results.

#### 2.5.1. Demographic Information

The students were surveyed based on their age, gender, study program, place of birth, and place in which they grew up. Regions were classified as urban or rural according to the 2012 Tanzanian census. If the birth region differed from the region in which they grew up, the latter was used in the classification as this was presumably where most of their experiences in local communities took place. In cases in which more than one region name was mentioned, the one that could be found in the 2012 Tanzanian census was chosen.

The regions were further classified into ecoregions on the basis of the climate cited in the Global Ecological Zones classification system of the Food and Agriculture Organization of the United Nations [15]. Participants who had grown up in more than one region or in a region outside Tanzania were excluded. Villages and wards that could not be found on the map or reasonably assigned an assumed climate were excluded from the ecoregion classification.

#### 2.5.2. Hospital Resources for Snakebite Treatment

Participants answered binary yes/no questions regarding the availability of hospital resources for snakebite treatment, namely, polyvalent antivenoms, intravenous drips, whole blood clotting tests, artificial respiration equipment, and Ministry of Health guidelines.

#### 2.5.3. Care of Snakebite Victim

Participants were asked to write down the duration, in hours, between each victim’s snakebite and arrival at the hospital. They also answered multiple-choice questions on what the victim did prior to arriving at the hospital. Multiple answers could be selected, and a qualitative open-ended section was offered for further explanations and elaboration of techniques that were not included. Binary yes/no questions on whether the victim survived and whether the victim suffered lasting damage were also included to assess treatment outcomes.

### 2.6. Qualitative Analysis

Participants wrote answers to open-ended questions in English and/or Kiswahili, and the answers in Kiswahili were translated. The answers were then entered into Excel tables, and content analysis was conducted to identify major themes.

Multiple themes and subthemes were identified from each participant and tabulated into codes. Their frequency was then calculated with RStudio 1.2.5033. As each theme was explored separately and each participant mentioned multiple, variable themes. The themes are not mutually exclusive, and the percentages of different themes thus do not add up to 100%. Subthemes, on the other hand, could be calculated as a percentage of their overarching theme, because they were mutually exclusive.

## 3. Results

### 3.1. Demographics

A total of 312 participants completed questionnaires from 2014 to 2019: 57 in 2014, 75 in 2015, 37 in 2016, 34 in 2017, 55 in 2018, and 54 in 2019. The vast majority of students (88%) enrolled in the eligible degree programs during that period consented to participate in the study.

Of the 312 students who participated in the survey regarding snakebite-related practices in their local communities, 71 participated in the subsequent sections on hospital resources for and outcomes of snakebite treatment. Two of the 71 participants were excluded because they explicitly mentioned treating a victim at home, leaving 69 participants.

Most of the participants were Bachelor of Nursing Students and between 20 and 25 years old; and slightly more than half of the participants were male. The majority of the participants had grown up in rural areas and in ecoregions such as tropical shrubland, which are large areas of land with seasonal rainfall covered by grasses rather than trees [16], or a tropical mountain system, which are mountainous areas with very high biodiversity and are important as a source of water [17] (see Table 1).

### 3.2. Community Practices to Treat Snakebite

Community practices were separated into two broad groups: allopathic and traditional. Allopathic practices were based on the 2010 WHO guidelines and the 2013 Tanzanian Ministry of Health and Welfare guidelines [18,19], which included taking the patient to the hospital, administering antivenom, and removing snake fangs from the wound. Other practices fit the WHO definition of traditional medicine—“the sum total of the knowledge, skill, and practices based on the theories, beliefs, and experiences indigenous to different cultures, whether explicable or not, used in the maintenance of health as well as in the prevention, diagnosis, improvement or treatment of physical and mental illness” [20]—and were classified as such.

Nearly all participants reported traditional practices (either alone or in combination with allopathic practices) (see Table 2).

The most common response to snakebite was to take the victim to a hospital, followed by the application of tourniquets. The response to snakebite often involved a combination of traditional practices in the following sequence: first attempting to remove the venom by incision with a blade, washing the wound, or mechanical suction with the mouth. Some participants also described the removal of snake fangs from the wound (See Figure 1).

Subsequently, community members typically applied a tight arterial tourniquet and then attempted to remove the venom using traditional objects that they believed absorb venom: snake stones (*jiwe la nyoka*) and their various permutations (dark stone, black stone, charcoal, etc.), as well as coins often in the form of 100- or 50-Shilling coins (13 and 2 participants, respectively).

In addition, about one-third of participants reported that community members sought help from traditional healers who typically used mixtures that included oral or topical administration of herbs and oral administration of milk. Three participants also mentioned the topical administration of ashes from dead snakes, two participants reported the topical administration of kerosine, and one participant mentioned the use of egg yolk and the oral administration of fresh dog ear.

Finally, the patient is taken to the hospital. Hospitalization is the only specific allopathic community practice that is used consistently, and only about half reported this practice (See Figure 2).

About one-tenth of participants reported variations of the response “provide first aid” without elaborating on the specific practices; these were classified as allopathic and were the next most frequently mentioned practice in this category. Several participants also reported the importance of antivenom as well as the necessity of identifying the snake.

Participants also reported other allopathic practices that the WHO recommends: immobilizing the patient to slow the spread of venom (two reports), and staying calm, maintaining hygiene, placing the injured part below heart level, avoiding tourniquets, using the pressure-immobilization technique, and finding the official treatment guidelines for further instructions (one report each).

### 3.3. Hospital Resources for Snakebite Treatment

Participants reported that about three-fourths of hospitals had intravenous drips upon arrival of a snakebite patient. Artificial respiration and whole blood clotting tests were each available in about half of the hospitals. Only about two-fifths of hospitals had the Ministry of Health guidelines for treating snakebite and polyvalent anti-snake venom (pAV) (See Table 3).

The five questions regarding hospital resources for snakebite treatment were left blank by more than half of the participants (53% left the question on intravenous drips blank and 55% left the questions on artificial respiration, Ministry of Health guidelines, pAVs, and whole blood clotting tests blank), presumably due to a lack of knowledge.

### 3.4. Care of Snakebite Victim

More than half of patients arrived at the hospital within 2 h of receiving a bite, about one-third between 3 and 6 h, and the rest took a day or longer. The median time between the bite and arrival at the hospital was 2 h.

Before arrival at the hospital, the bite site was most frequently treated with a tourniquet (67%) and/or snake stone (59%), followed by washing (31%), incision (28%) and/or herbal treatment (23%). Less commonly, patients came with no treatment or had applied dung. A significant number of patients (20%) also used treatments not included in the multiple-choice options: four removed snake fangs from the wound; two patients each administered an analgesic orally, applied kerosine and bound the wound; and one patient administered milk orally and identified the bite site and attempted mechanical suction of the venom.

More than one in ten snakebite patients died from the envenoming, and about one in three suffered lasting damage (see Table 4). This included two patients each with amputations and scarring from necrosis, and one patient with psychological trauma and injury during the removal of snake fangs. One participant reported that a snakebite victim died because the nearest hospital was too far away and another one reported a death following referral to another hospital for antivenom.

## 4. Discussion

We gathered information on local community practices and hospital resources to treat snakebite from nursing students at one of East Africa’s premier institutions of nursing and medical education, the Muhimbili University of Health and Allied Sciences (MUHAS). The majority of our study participants were enrolled in the Bachelor of Science in Nursing program. In total, there are ten universities in Tanzania offering this course [21], and the students in these institutions tend to be similar to the students in our study. At the same time, the students in our study came from various regions in Tanzania, so it is likely that our results will generalize well to the experiences of nursing students with snakebite at the national level in Tanzania. We specifically asked for any experiences with snakebite, including treatments that the students had administered themselves or merely observed, which likely further boosts the generalizability of our findings. The students’ answers may nonetheless be distorted because they reflect the experiences of highly educated individuals in Tanzania, where only about one in six people attends upper secondary [22], and the students in our survey are of course also specifically trained in healthcare, which may affect where, when, and how they experience the activities and contexts of snakebite treatment.

### 4.1. Community Practices to Treat Snakebite

The most common community practice to treat snakebite reported in our study was the tight arterial tourniquet. This practice is widespread throughout Africa among both healthcare professionals and the general population [23]. It is believed to limit the spread of venom to the rest of the body; however, applying a tourniquet concentrates the venom in the affected limb, which increases local tissue damage [24] in the form of nerve injury, ischemia, and gangrenous limbs. More importantly, release of the tourniquet may cause shock, pulmonary embolism, and death due to venom flooding [19]. One alternative method is the pressure-immobilization technique, which was reported once in our study. This method originated in Australia and involves stabilizing the affected limb with a splint to prevent movement, and wrapping a bandage tightly enough that it restricts lymphatic flow but not blood flow [25]. This technique, however, requires high expertise and quality bandages to exert the correct amount of pressure (40–70 mmHg in the upper extremity and 55–70 mmHg in the lower extremity) and is not recommended for use by a layperson [26,27]. Finally, the WHO-recommended approach—to immobilize the affected limb and to avoid movement to slow the spread of venom [19]—was only reported once in our study. Overall, the community practices intended to reduce the spread of the venom demonstrate that snakebite treatment could be substantially improved by interventions that boost snakebite treatment knowledge and skills to ensure that the harmful practice of applying tourniquets is replaced with the WHO-recommend immobilization approach.

Our study participants further reported a wide range of practices intended to remove venom: incising; washing; and applying suction, snake stones (*jiwe la nyoka*), and Shilling coins. Incising, washing, and suction were recommended a few decades ago. Their efficacy has since been disproven [19,28]. For instance, suction devices such as the Extractor removed bodily fluid but no venom [28]. These methods are discouraged by the WHO and may damage tendons, nerves, and arteries; facilitate absorption of the venom; and introduce infection and cause increased swelling [19,29]. 

Snake stones have existed since antiquity and are extensively used not only in Tanzania but also throughout Africa, Asia, and parts of Latin America in varying forms [30]. In Tanzania, they are chiefly palm-sized, round to oval-shaped, dark-colored stones that are applied to the bite site. According to traditional belief, their tight adherence to the surface of the bite site helps to draw out venom, and boiling the stones in milk regenerates their healing properties [31]. In Tanzania, Shilling coins are believed to have similar absorbent properties [28,32]. Like mechanical suction, snake stones and Shilling coins are likely not efficacious, and indeed studies have shown that venom spreads too quickly or is injected too deeply to be meaningfully extracted in any manner [28,32]. The WHO guidelines recommend against using any venom suction or extraction methods, and other studies, including one which specifically tested the stone against the venom of snake species commonly found in Africa (*Bitis arietans*, *Echis ocellatus*, *Naja nigricollis*), have also shown that they are ineffective in removing venom from the bite site [24,31]. However, they are shown to cause no additional death or disability [24]. Snake stones and Shilling coins may help calm snakebite victims as long as these applications do not delay treatment in a hospital.

About one-third of the participants in this study reported turning to traditional healers for help. Traditional healers treated snakebite mainly with herbs, which play a central role in African traditional medicine. The specific types and amounts of herbs used are often not well known, in part because of the secrecy and competition surrounding traditional healing [33]. Consequently, herbal snakebite treatments are not standardized and tend to vary among the numerous types of traditional healers. Indeed, ‘traditional healer’ and ‘traditional medicine’ are terms that have been criticized as ethnocentric because they group together extremely diverse groups of practitioners whose main commonality is that they provide diagnoses and treatments that are not allopathic [33,34]. Cooperation between traditional and allopathic health workers could improve snakebite treatment pathways and outcomes because snakebite patients and their relatives often see traditional healers before they seek help from allopathic health workers.

Most snakebite victims do not require hospitalization—less than 10% of the 3500 snake species worldwide are venomous, and the rate of ‘dry bites’ (bites without the injection of venom) ranges from about 23% to 50% [18,19,35]. However, in the absence of an accurate triage system, timely hospital visits are critical for snakebite outcomes because “no first-aid method [is] effective against all snake bites” [32]. Most snakebite victims in the present study were reported to have arrived at a hospital within six hours of being bitten; however, 14% took a day or longer to arrive. The precise reasons for the delays, including the distance from the location of the snakebite to the nearest hospital, are unclear. Considering, however, that about one-fourth of victims had herbs or dung applied to their bite site prior to their arrival, and even more were treated with tourniquets and snake stones, it is likely that community and traditional treatments contributed significantly to delayed arrivals at the hospital. Again, interventions that boost community knowledge and skills to treat snakebite, as well as strengthened cooperation between traditional healers and hospitals, could likely reduce the time from snakebite to adequate diagnosis and treatment in hospitals. Traditional healers have shown great willingness to collaborate more effectively with hospital staff and sometimes refer patients to hospitals [36], suggesting that cooperation between traditional healers and allopathic institutions to improve the treatment pathways and outcomes following a snakebite are indeed possible [36].

A systematic review has shown that ingesting or topically applying traditional herbal treatments of snakebite causes a statistically significant increase in death or disability, and no decrease in the duration of hospital stays [27]. Other concerns regarding traditional herbal treatments are poor quality control, lack of documentation during the treatment, unknown side effects, and adverse interactions with other substances [36]. Overall, however, our understanding of commonly used traditional herbal treatments for snakebite is very rudimentary, and in-depth studies should elucidate which herbs may be valuable for treating snakebite [37].

### 4.2. Hospital Resources for Snakebite Treatment

Antivenom works by binding to and neutralizing snake venom through the antigen-antibody reaction [38]. It is currently the only specific treatment available for SBE. Yet, the nursing students in our study reported that antivenom was mostly not available for snakebite patients in hospitals. One important reason for the lack of antivenom in Tanzanian hospitals may be its high price (ranging from USD 55 to 640 per effective treatment) [13,39], which is due to the laborious and expensive antivenom production process: Snake venom has to be extracted by skilled handlers, each venom type injected into different equines or bovines, and the resulting antibodies properly purified to reduce the risk of anaphylactic shock [7]. In addition, very few companies produce high-quality antivenom specifically for the African market, and many of the products that are commercially available in Africa lack clinical efficacy and safety data [40]. It is thus unsurprising that many hospitals do not stock sufficient antivenom to treat all snakebite patients. Another factor that likely complicates antivenom availability in hospitals is the lack of reliable cold storage (antivenom has to be stored at 2–8 °C [41]). Finally, snakebite incidence varies widely over time and underreporting is common, making it difficult to plan the purchasing and stocking of antivenom to ensure sufficient supply with minimum wastage.

The case of the nurse-led Meserani Snake Park clinic, which specializes in snakebite treatment, demonstrates the snakebite outcomes that can be achieved when antivenom is reliably available. This clinic is the only one in Tanzania that stocks polyvalent antivenom specifically targeted at snakes in Tanzania. The number of patients increased steadily from 2007 to 2009, with people traveling an average of 82 km for treatment [42]. The clinic is funded by the income of the associated snake park as well as by donations, and medical care at the clinic is completely free of charge. The outcomes of snakebite treatment are reported to be relatively good. A study found that only 1% of the snakebite victims treated at the Meserani Snake Park died, and only 7% required a skin graft or amputation [42]. These rates are much lower than those reported by the participants in our study: about one in six snakebite victims did not survive and about one in three suffered long-term damage. The Meserani Snake Park clinic may be useful as a model to boost snakebite treatment in other locations in Tanzania, including within existing allopathic hospitals. 

We also found major gaps in supportive care items—intravenous drips, artificial respiration, and whole blood clotting tests—that are necessary to successfully treat snakebite from several snakes that are common in Tanzania. Artificial respiration is essential in cases of exposure to snakes with neurotoxic components such as mambas (*Dendroaspis angusticeps* and *Dendroaspis polylepis*) and puff adders (*Bitis arietans*), where SBE results in gradual muscle paralysis, eventually leading to respiratory failure. Whole blood clotting tests are a useful diagnostic tool for venom that contains hematotoxins from snakes such as vipers (*Echis pyramidum*, *Bitis gabonica* and *Bitis nasicornis*), which cause extensive bleeding [8,19]. These tests assess the coagulation mechanism and can indirectly detect the presence of hematotoxins in the blood, which help in snake identification and in predicting the severity of SBE [43,44]. Initiatives to boost the availability of these supportive care items would likely reduce the number of deaths from snakebite in Tanzania (as well as support care for many other conditions).

While intravenous drips, artificial respiration, and whole blood clotting tests are costly and need to be replenished after use, another important item supporting snakebite treatment, the *Standard Treatment Guidelines and Essential Medicines List* [18], is inexpensive and can be used repeatedly over long periods of time. The Tanzanian Ministry for Health and Social Welfare has developed this document in accordance with WHO recommendations, and it includes specific guidance on treating snakebite. Our study suggests that this document is typically not available when a snakebite is being treated. There is also general evidence that the use of the guidelines could boost quality of care in Tanzania [45]. Ensuring the consistent availability of snakebite guidelines in healthcare facilities in Tanzania is likely a feasible and affordable intervention to boost the success of snakebite treatment (and many other treatments).

## 5. Conclusions

Our study revealed three major findings regarding snakebite community practices and hospital resources in Tanzania, all of which led to intervention recommendations. First, snakebite community practices in Tanzania are largely ineffective and often harmful. Community education and snakebite first-aid skills training can thus likely improve snakebite outcomes in Tanzania. Second, snakebite victims in Tanzania often see traditional healers before they see allopathic health workers, suggesting that collaboration between the traditional and the allopathic health system in Tanzania could be leveraged to boost snakebite outcomes. Third, antivenom and other important resources to treat snakebite are commonly missing in Tanzanian hospitals. Future initiatives to ensure consistent availability of these resources—in particular, polyvalent antivenom specifically targeted to snakes in Tanzania—would likely boost survival among snakebite victims and reduce the incidence of long-term damage. 

## Figures and Tables

**Figure 1 ijerph-19-04701-f001:**
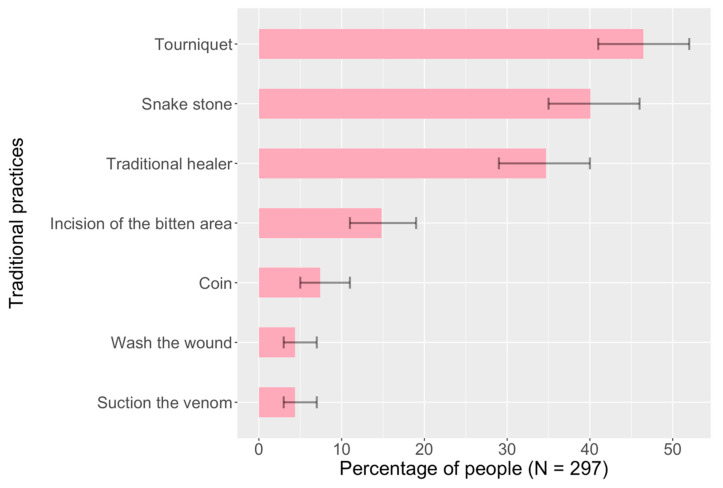
Traditional community practices to treat snakebite.

**Figure 2 ijerph-19-04701-f002:**
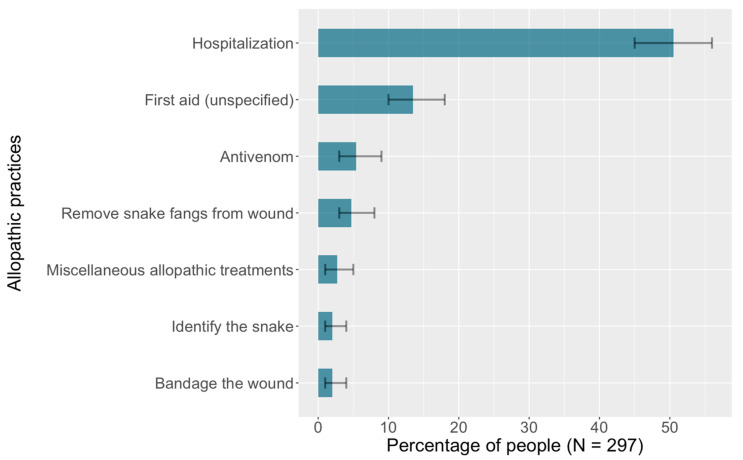
Allopathic community practices to treat snakebite.

**Table 1 ijerph-19-04701-t001:** Demographic characteristics of the participants.

Factors	Level	% (95% CI) *
Age (years)	20–25 26–50	78 (73–82) 22 (18–27)
Sex	Male Female	60 (54–65) 40 (35–46)
Location	Rural Urban	78 (73–83) 22 (17–27)
	Tropical dry forest Tropical moist deciduous forest Tropical mountain system Tropical rainforest Tropical shrubland	10 (7–14) 16 (12–21) 24 (20–29) 15 (11–20) 35 (29–40)
Program	Bachelor of Science in Nursing Other	80 (75–84) 20 (16–25)

CI = confidence interval. * N = 306 for age. N = 309 for sex and location (rural/urban). N = 291 for location (ecoregion). N = 308 for program.

**Table 2 ijerph-19-04701-t002:** Community practices to treat snakebite by category (N = 297).

Community Practices	% (95% CI)
Only traditional	44 (39–50)
Only allopathic	7 (5–11)
Both	49 (43–54)

CI = confidence interval.

**Table 3 ijerph-19-04701-t003:** Hospital resources for snakebite treatment.

Factors	Level	% (95% CI) **
Was an intravenous drip available?	Yes No	75 (58–87) 25 (13–42)
Was artificial respiration available?	Yes No	55 (38–71) 45 (29–62)
Were Ministry of Health guidelines for treating snakebite available?	Yes No	42 (26–59) 58 (41–74)
Was polyvalent anti-snake venom available?	Yes No	42 (26–59) 58 (41–74)
Was the whole blood clotting test available?	Yes No	48 (32–65) 52 (35–68)

CI = confidence interval. ** N = 31 for artificial respiration, Ministry of Health guidelines, polyvalent anti-snake venom, whole blood clotting test; N = 32 for intravenous drip.

**Table 4 ijerph-19-04701-t004:** Characteristics of snakebite victims upon arrival at the hospital and hospital outcomes (N = 69).

Factor	Level	% (95% CI)
How many hours before the patient arrived at the hospital for treatment? (N = 41)	<1 h 1–2 h 3–4 h 5–6 h 1 day or more	27 (16–42) 29 (18–44) 15 (7–28) 15 (7–28) 14 (7–28)
Which of the following was done (before arrival to the hospital)? (N = 64)	a. No treatment b. Washing the wound c. Applying tourniquet d. Cutting the site of the snakebite e. Applying dung f. Applying herbal treatment g. Applying snake stone h. Other (explain) Removal of snake fangs Other answers *	6 (2–15) 31 (21–43) 67 (55–77) 28 (19–40) 5 (2–13) 23 (15–35) 59 (47–71) 20 (12–32) 6 (2–15) 14 (8–25)
Did the patient die? (N = 58)	Yes No	14 (7–25) 86 (75–93)
Was the patient left with lasting damage? (N = 41)	Yes No	32 (20–47) 68 (53–80)

CI = confidence interval. * Other answers are available in the text above.

## Data Availability

The data presented in this study are available in the Appendix A.

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
