# Peer review of "Snakebite Treatment in Tanzania: Identifying Gaps in Community Practices and Hospital Resources"

_ijerph, 2022, doi:10.3390/ijerph19084701_

Round 1

Reviewer 1 Report

Thank you for the opportunity to review this cross-sectional mixed methods study of student health professionals' knowledge and opinions regarding snakebite care. Please consider the following comments/suggestions:

Introduction: the introduction is too long. It discusses important issues, but does not focus well on laying out a brief background and demonstrating why this specific research question is important. Much of the intro would be better served in the discussion or removed altogether as superfluous to the main research question. The introduction should end with a sentence regarding the purpose or objective of the study.

Lines 51-63 could easily be removed and the manuscript would not suffer. 

Lines 64 - 80. This could be shortened dramatically and focused towards the fact that antivenom is the foundation of treatment, yet unproven traditional therapies are common and this is one reason the research question is important.

Lines 81-85. Either remove or incorporate early into the background information.

Materials and Methods:

Line 92-95. Much of this sentence belongs in the last sentence of the introduction. The study design itself should include the term mixed methods as there are quantitative and qualitative elements.

Lines 108-113. The description of the students belongs in this are, not in "demographic information". As student training, clinician roles, etc. vary all over the world, a description of this participants in the context of healthcare delivery in Tanzania is vital.

lines 136-138. It seems that the place they have worked would be more important than n the place they grew up. However, maybe these are almost always the same thing. I would just clarify why "the place they grew up" is so impactful to their professional opinions/approach. That certainly is not universally the case globally.

Results:

Lines 251-252. The data is non-parametric and you should report the median, not the mean. I will note that it seems unusual that there is a gap between 6 hrs and 24 hrs where nobody presented.

Discussion: The discussion is also too long and reads like a review paper on inappropriate snakebite therapy. It should focus on what beliefs these student health professionals had and use existing literature to discuss/propose reasons as to why they have them. the downside of these therapies can be discussed, but not in this degree of detail. The current two to three paragraphs on each ineffective therapy does not allow the authors to paint a coherent picture as to why these beliefs are held and what can be done about them.

4.3.1. Polyvalent Anti-Snake Venom. this section should succinctly demonstrate that antivenom is the backbone of current snakebite therapy, but that challenges in access may lead to these ineffective other therapies. Basically, keep lines 399-411.

Lines 377-391. Most of this can removed and just briefly explain why expensive. 

Supportive care. Most of this section should be removed as it reads like a review or care manual. This is not the point of this study's results. 

Conclusions: Keep the first paragraph as a conclusion of the entire study. The subsequent two paragraphs are really more appropriate for the discussion section. The reader need a brief tightly written recommendation that is directly tied to the results of the study. 

Reviewer 2 Report

The paper is a descriptive works, which shows the real situation of snake envenomation’s treatment in Tanzania. All results were collected by health workers who had experience with snake bitte. The authors demonstrated the reality of snake envenomation’s treatment in Tanzania. The paper also showed that some practices strongly disapproved by WHO are widely used nowadays in regions of Africa, as Tanzania. The authors highlight how snake envenommation is neglected in some regions of the world, further reinforcing the scientific advance to offer quality and affordable treatments. I congratulate the authors for showing how snakebite remains neglected and how efficient treatment should be offered for population around the world. I also would like suggest some modifications to improve the paper’s quality.  

  • The methods were well designed; however, the authors did not show any statistical analysis. Please provide the analysis, describe and include these on methods section;
  • Line 173: Please fix the date of year;
  • Line 181: Explain the term “shrubland”. This information is important to understand the geographical context of snake envenomation in Tanzania;
  • Line 228: Please, fix the term “poison”, the correct word is “venom”;
  • Table 3 and Figures 1 and 2 have the same information. Please, choose just one way to show this data;
  • Figure 3: Please, refine the figure’s design;
  • The authors must compare the data from health workers with epidemiological data from official organization in Tanzania;
  • The authors must show more information about the biology of snakes responsible for the envenomation reports in Tanzania, as well as add more information about clinical data of snake envenomation in the country.

Best regards.

Round 2

Reviewer 1 Report

Thank you for incorporating my prior suggestions into the revised manuscript. You have addressed my primary concerns. I have one suggestions regarding the revised conclusion paragraph. This study has highlighted 3 areas of deficit in snakebite management in Tanzania: 1. clinician education 2. supportive care 3. antivenom. I think you could more explicitly state this in the conclusion with minor changes. Consider the following minor changes to your conclusion:

First-aid intervention in local communities often do not follow WHO recommendations, due to lack of clinician education, necessary resources to provide supportive care, and polyvalent anti-snake venom. Hospital staff should be trained to handle snakebite injuries effectively and in accordance with recommended protocols. Medical facilities should also improve the quality of supportive care resources and available pAVs. This will improve outcomes and likely encourage more people to seek available guideline-based care.

Author Response

Thank you very much for your excellent suggestions, we have made the changes to the conclusion accordingly.

Additional notes:
- In refining the conclusion, we also noticed that ‘hospital resources’ appear to be more widely used than ‘hospital facilities’ (which may refer to simply the building), and also that ‘community practices to treat snakebite’ may make more sense than ‘first-aid methods’ in this context.
- Figures 1 and 2 were reordered to standardise traditional first and then allopathic.
- We also improved the language and overall clarity of the paper, especially in the discussion section.